# Single-Cell RNA Sequencing of Donor-Reactive T Cells Reveals Role of Apoptosis in Donor-Specific Hyporesponsiveness of Kidney Transplant Recipients

**DOI:** 10.3390/ijms241914463

**Published:** 2023-09-23

**Authors:** Amy C. J. van der List, Nicolle H. R. Litjens, Rutger W. W. Brouwer, Mariska Klepper, Alexander T. den Dekker, Wilfred F. J. van Ijcken, Michiel G. H. Betjes

**Affiliations:** 1Erasmus MC Transplant Institute, Department of Internal Medicine, University Medical Center, 3015 CN Rotterdam, The Netherlands; a.vanderlist@erasmusmc.nl (A.C.J.v.d.L.); n.litjens@erasmusmc.nl (N.H.R.L.); m.klepper@erasmusmc.nl (M.K.); 2Erasmus MC Center for Biomics, University Medical Center, 3015 CN Rotterdam, The Netherlands; r.w.w.brouwer@gmail.com (R.W.W.B.); a.dendekker@erasmusmc.nl (A.T.d.D.); w.vanijcken@erasmusmc.nl (W.F.J.v.I.)

**Keywords:** donor-reactive, alloreactive, T lymphocyte, hyporesponsive, flow cytometry, clustering, single cell, expression profiling

## Abstract

After kidney transplantation (KT), donor-specific hyporesponsiveness (DSH) of recipient T cells develops over time. Recently, apoptosis was identified as a possible underlying mechanism. In this study, both transcriptomic profiles and complete V(D)J variable regions of TR transcripts from individual alloreactive T cells of kidney transplant recipients were determined with single-cell RNA sequencing. Alloreactive T cells were identified by CD137 expression after stimulation of peripheral blood mononuclear cells obtained from KT recipients (N = 7) prior to and 3–5 years after transplantation with cells of their donor or a third party control. The alloreactive T cells were sorted, sequenced and the transcriptome and T cell receptor profiles were analyzed using unsupervised clustering. Alloreactive T cells retain a highly polyclonal T Cell Receptor Alpha/Beta repertoire over time. Post transplantation, donor-reactive CD4+ T cells had a specific downregulation of genes involved in T cell cytokine-mediated pathways and apoptosis. The CD8+ donor-reactive T cell profile did not change significantly over time. Single-cell expression profiling shows that activated and pro-apoptotic donor-reactive CD4+ T cell clones are preferentially lost after transplantation in stable kidney transplant recipients.

## 1. Introduction

Following kidney transplantation, a gradual decrease in the reactivity of recipient T cells to donor antigen, termed donor-specific hyporesponsiveness (DSH), is observed in some patients [1,2,3,4,5,6]. A better understanding of the mechanisms underlying the development of DSH in T cells could guide the lowering of immunosuppressant medication, thereby reducing the risk of unwanted long-term side effects such as infection, cancer and cardiovascular disease, which are especially prevalent in elderly kidney transplant recipients [7,8].

Previous studies have explored T cell regulation, anergy, exhaustion and clonal deletion as possible mechanisms leading to DSH development [9,10,11,12]. Recently, data collected in our group with CD137-based assays support a role for apoptosis of donor-reactive CD4+ memory T cells [13,14]. A limitation of these studies was reliance on a selection of read-outs to characterize the highly heterogeneous donor-reactive T cells. Next generation sequencing (NGS) methods, with their ability to discern the transcriptional differences between cells with a unique T cell receptor (TCR), could help unravel the complex molecular mechanisms of DSH development.

Several groups have designed an assay to sequence the TCR repertoire of donor-reactive T cells within PBMCs of transplant recipients [15,16,17]. In these assays, donor-reactive T cell clones were defined as those clones which expanded from a sequenced population of either CD154-expressing or -proliferating T cells, following a mixed lymphocyte reaction (MLR) of recipient T cells with donor antigen [15,16,17]. Using these assays, researchers have provided evidence that the deletion of donor-reactive T cells may be the leading cause of DSH within tolerant combined kidney and bone marrow and liver transplantation [15,18]. However, these studies were limited to bulk TCR sequencing (mainly focused on TCRβ chain profiling) and did not measure the transcriptome of individual donor-reactive T cells. 

Directly linking TCR sequencing with the transcriptome at the single-cell level would allow for the observation of changes in the transcriptional profiles of specific T cell clones post-transplantation. The introduction of VDJ target enrichment combined with single-cell sequencing platforms has enabled T Cell Receptor Alpha/Beta (TRA/TRB) chain sequence pairing and the integration of clonality information with the whole transcriptome of a single T cell [19]. Frequencies of alloreactive T cells are very low, which provides a challenge for single-cell sequencing, especially when limited by the amount of patient material available. Recently, we have validated an assay for the sorting of low numbers of T cells of interest and a technique to measure both their transcriptome and TCR alpha beta repertoire [20,21]. 

In this study, an analysis of the transcriptome and TRA/TRB repertoire of donor-reactive T cells was performed to elucidate mechanisms leading to donor-specific hyporesponsiveness. 

## 2. Results

### 2.1. UMAP Clustering of Donor-Reactive T Cells

Clustering the sequenced cells based on transcriptome (RNA expression) resulted in a total of four clusters (clusters 0, 1, 2 and 3), as depicted in a UMAP plot (Figure 1A). T cell marker genes CD3D encoding the delta chain of the CD3 protein and CD137 (*TNFRSF9*) were expressed in all clusters (Appendix A). Non-T cell markers including *CD14* expressed by monocytes, *CD19* indicating B-lymphocytes and NCAM1 (*CD56*) encoding for natural killer cells were not expressed in clusters 0 to 2 (Appendix A). This expression pattern confirms that the cells sequenced were indeed the CD137+ donor-reactive T cell population of interest. Expression of *CD8A* and *CD8B* (encoding the alpha and beta-chain of CD8+ protein) is concentrated in cluster 2, while expression of *CD4* is present in clusters 0 and 1 (Appendix A). The final cluster, cluster 3, contained a very limited number of cells and showed a significant upregulation of *CD14* expression (adjusted *p*-value of 8.79 × 10^−13^ and log2FC of 3.48) (Appendix A). Cluster 3 was therefore excluded from further analysis.

A greater proportion of alloreactive T cells sequenced were CD4+ T cells. Unpublished data from a previous study using the same cohort of stable kidney transplant recipients, characterizing alloreactive T cells at the protein level, revealed that the majority of alloreactive (CD137+) CD3+ T cells were CD4+ T cells and this was not affected in time, i.e., comparing post-transplant to pre-transplant. 

### 2.2. Differential Expression Analysis Determined T Cell Gene Expression Profile per Cluster

Differential expression analysis between clusters was used to define the cell identity and function of each cluster (Figure 1B). Cluster 2 contained cytotoxic CD8+ T cells, as evidenced by a significantly higher expression of CD8+ T cell markers (*CD8A, CD8B*), cytotoxic genes (*GNLY, GZMB, NKG7, KLRD1, GZMH*) and genes encoding for chemokines (*CCL4L2, CCL5, CCL4, CCL3*) compared to other clusters (Appendix A). 

Within clusters 0 and 1 containing alloreactive CD4+ T cells, a few notable genes involved in immunity and cytokine-driven bioprocesses were inversely regulated. *AC124319.1* (*RNF213*), *CD74*, *RGS1*, *LGALS1* and *IL32* were upregulated in cluster 0 but downregulated in cluster 1 (Appendix A). In addition, cluster 0 had upregulation of *TIGIT* (marker for activation and exhaustion of T cells), as well as *ISG20* and *LTB* (involved in interferon signaling and inducing the inflammatory response, respectively). Cluster 1 had upregulation of *IL2*, *HSPE1*, *FABP5*, *NOP16*, *NPM1*, *NME1*, *SLIRP*, *RAN* and *TOMM5* (Appendix A). Conversely, cluster 0 downregulated all of these genes with an absolute log fold change above 1, except for *RAN* which plays an important regulatory function in T cell activation. Notably, interleukin-2 (IL-2) is a cytokine produced by activated CD4+ T cells that plays pivotal roles in the immune response including proliferation, differentiation and cell survival/apoptosis.

None of the differential genes allowed us to distinguish specific T cell subtypes such as naïve, memory or regulatory. Instead, these genes are expressed across all clusters (Appendix A). 

### 2.3. Gene Ontology Pathways Defined Cell Activity per Cluster

To further decipher the function of the cells within each cluster, we performed a Gene Ontology (GO) term analysis on all differentially expressed genes (adjusted *p*-value < 0.01) to determine which pathways these genes up- or downregulated (Figure 1D; Appendix A). 

GO Term analysis results for cluster 2 are in line with the cells being cytotoxic T cells with upregulation of ‘immune system/effector process’ (GO:0002376, GO:0002252) and upregulation of ‘defense/immune response’ (GO:0006952, GO:0006955). For clusters 0 and 1, an inverse relationship was found. Cluster 0 was enriched for immune response including ‘response to external biotic stimulus’ (GO:0043207) and ‘immune system process’ (GO:0002376). Inversely, cluster 1 had downregulation for immune system process and immune response pathways.

In summary, the GO term analysis confirms the presence of three major clusters of T cells. One cluster (2) had a profile consistent with activated, cytotoxic donor-reactive CD8+ T cells, and the other 2 clusters (0 and 1) contained donor-reactive CD4+ T cells with differing degrees of activation and pro-inflammatory status.

### 2.4. Differential Expression and GO Term Analysis between Post and Pre-Transplant Donor-stimulated Samples within Each Cluster

Gene expressions within each cluster of cells originating from either a post or pre-transplant donor-stimulated sample were compared to determine which genes are differentially expressed 3–5 years post kidney transplantation. Table 1 and Table 2 summarize all genes reaching an adjusted *p*-value below 0.01 between samples from a post versus a pre-timepoint within cluster 0 and 1, respectively. 

Cluster 0 (TIGIT-expressing CD4+ T cells) showed significant downregulation of *TYMP* and *PMAIP1* post-transplantation, both genes known to play a role in apoptosis (Table 1; Figure 2A). Downregulation of RNF213 was also observed with a p-adjusted value of *p* < 0.05. In conclusion, gene expression changes from before to 3–5 years after transplantation in the TIGIT-expressing donor-reactive CD4+ T cells (cluster 0) show a downregulation in expression of apoptosis-related genes.

For cluster 1 (IL2-expressing CD4+ T cells), we observed a downregulation of genes known to play a role in activation, proliferation and/or apoptosis in donor-reactive T cells from post-transplant timepoints. In this cluster, this was evidenced by decreased expression of *EIF5A, PA2G4, IL15RA, TNFRSF25, GADD45A, TAPBP, BZW2, NFKBIA* and *CYS* (Table 2; Figure 2B). 

Of note, no differences in gene expression between cells from post versus pre-transplantation were observed within cluster 2 containing the CD8+ T cells. 

Next, the pathways affected by the differentially expressed genes between pre- and post timepoints were analyzed with GO Term analysis. The number of differentially expressed genes within cluster 0 was insufficient for GO Term analysis. Within cluster 1, when considering all genes differentially expressed post-transplant (adjusted *p*-value < 0.1), we observe the downregulation of cytokine-mediated signaling pathways, including those mediated by tumor necrosis factors (GO:0019221, GO:0033209) and peptidyl-serine phosphorylation of STAT protein (GO:0033140, GO:0033139, GO:0042501, GO:0033137) (Table 3). Notably, the genes differentially regulated post-transplant within cluster 1 were also involved in the downregulation of apoptotic processes (GO:0006915, GO:2001244, GO:2001242) (Table 3). 

As stated in the methods, we used a gene expression analysis of so-called third party alloreactive T cells as a control for donor kidney-unrelated effects of time after transplantation (e.g., use of immunosuppressive drugs). None of the genes differentially expressed from pre- to post timepoints following third-party stimulation were shared with donor-stimulated samples (Appendix A). Consequently, the related GO Terms did not overlap with those from donor-reactive T cells (Appendix A)

In summary, the changes in gene expression patterns observed in the donor-reactive T cells after kidney transplantation are specifically related to the presence of the donor organ and only found in the clusters containing CD4+ T cells. We observed a consistent decrease in cytokine-mediated signaling and apoptotic processes in these donor-reactive T cells several years after transplantation. 

### 2.5. Diversity of Donor-Reactive TCR Repertoire Does Not Change over Time

A high number of unique donor-reactive T cell clones were identified in both pre- and post-transplant samples (Figure 3). Donor-reactive T cells remained polyclonal post transplantation with the Shannon Equitability Index remaining unchanged at an average of 0.97 both pre-transplant and post-transplant (Table 4). This illustrates the high heterogeneity of donor-reactive T cells and no donor-specific change within the alloreactive T cell clonal diversity was observed over time.

## 3. Discussion

In this study, unsupervised clustering of single cell RNA sequencing data was used to compare both the transcriptome and TCR repertoire of donor-reactive T cells in stable kidney transplant recipients before and 3–5 years after kidney transplantation. Clustering based on the transcriptome divided the alloreactive T cells into three main groups: one cluster of cytotoxic CD8+ T cells and two clusters of CD4+ T cells with distinct activation profiles. One cluster of alloreactive CD4+ T cells was characterized by upregulation of TIGIT (cluster 0), while the other by upregulation of IL-2 (cluster 1). Other notable genes which were differentially expressed between cluster 0 and 1 included *RGS1*, *LGALS1* and *IL32* due to their relation to apoptosis. *LGALS1* and *IL32* are strong inducers of activation-induced cell death (AICD), with *IL32* specifically expressed in T cells undergoing apoptosis [22].

Differential expression analysis from pre- to post-transplantation revealed that donor-reactive CD4+ T cells within both clusters (cluster 0 and 1) had a downregulation of genes involved in apoptosis and T cell activation post-transplant. Notably, the genes significantly downregulated in cluster 0 (TIGIT-expressing CD4+ T cells), *TYMP*, *PMAIP1* and *RNF213,* are known to play a role in apoptosis. *PMAIP1* belongs to a pro-apoptotic subfamily within the BCL-2 protein family, referred to as the BCL-2 homology domain 3 (BH3)-only subfamily, which determines whether a cell commits to apoptosis [23]. *TYMP* is contained in multiple apoptosis pathways, including ‘p52 mediated apoptosis’, and multiple studies indicate its role in inhibiting apoptosis [24], and mutations in the RING domain of RNF213 have been shown to promote apoptosis [25].

Indeed, GO Term analyses revealed that the differential gene expression between donor-reactive CD4+ T cells pre- to post-transplant was associated with the downregulation of apoptosis and intracellular signaling pathways. Remarkably, no change in the transcriptome of donor-reactive cytotoxic CD8+ T cells was observed over time. The inclusion of third-party controls enabled us to ascertain that the differences we detected post-transplant were truly donor-specific and not due to the influence of immunosuppression. 

Tracking T cells based on T cell receptor TRA/TRB sequence pairing demonstrated that a high number of unique alloreactive T cell clones are present pre-transplant. We did not observe a decrease in the high TCR diversity of alloreactive T cell clones post-transplantation. The high diversity in alloreactive T cell TCR clonotypes both before and after transplantation has been shown in previous studies [17,26,27]. Due to almost every clonotype being unique, we did not find sufficient cells with overlapping TRA or TRB clonotypes between timepoints of the same patient to enable a transcriptome analysis for specific TCR clones (Appendix A). The low number of cells with shared clonotypes between samples (pre vs. post or donor vs. third-party) limited the conclusions which can be drawn in respect to TCR clonality.

Similar to previous studies by our laboratory and others, we found no evidence to support a role for increased T cell regulation or exhaustion within donor-reactive T cells post transplantation. [10,13] Instead, this work provides evidence on the RNA level in support of the hypothesis that specific apoptosis of donor-reactive CD4+ T cells drives DSH development within the first 3–5 years post-transplantation. The observed downregulation of genes and pathways involved in T cell activation post transplantation is in accordance with previous data from our lab obtained by flow cytometry. In these studies, a loss of highly activated donor-reactive CD4+ T cells 3–5 years post transplantation secreting multiple pro-inflammatory cytokines was observed [13,14]. In addition, increased susceptibility to apoptosis was associated with the preferential loss of these donor-reactive CD4+ T cells from the periphery [14]. However, the data obtained in this study show a decrease in pro-apoptotic pathways post transplantation, which seems to be in contradiction with an apoptosis-mediated loss of alloreactive T cells. This probably represents a paradox as the most activated alloreactive T cells are more prone to apoptosis and, thus, the cells remaining several years after transplantation show both a less activated and decreased pro-apoptotic profile. 

Interestingly, our previous studies did not observe a decline in donor-reactive CD8+ T cells until more than 10 years post-transplant [13,14]. Neither did we find indications for increased apoptosis among donor-reactive CD8+ T cells [14]. This coincides with the current study where no differences in the transcriptome of donor-reactive CD8+ T cells 3–5 years post-transplant were observed. 

This is the first study to investigate both TCR and transcriptome sequencing data of donor-reactive T cells. Groups that have sequenced donor-reactive T cells in transplantation have undergone bulk sequencing and/or focused on the TCR sequence only without consideration of the transcriptome [28]. These studies similarly observed that deletion of certain donor-reactive T cells likely underlies the development of DSH. Morris et al. (2015) defined the anti-donor TRB repertoire by sequencing pre-transplant recipient T cells which proliferated in response to donor stimulation in a mixed lymphocyte reaction (MLR). The authors demonstrated that tolerant combined kidney and bone marrow transplantation patients had a significant reduction over time in the number of circulating donor-reactive clones compared to pre-transplant. This finding did not coincide with observations in conventional kidney transplant recipients, which could be due to the low number of samples (N = 2) [15]. Overall, this study provided evidence for the role of deletion in the maintenance of allograft tolerance. This same anti-donor TRB sequencing technique was used by Savage et al. (2020) to obtain evidence consistent with the deletion of donor-reactive T cells post transplantation in both tolerant (N = 3) and non-tolerant (N = 5) liver transplant recipients [18]. Similarly, Aschauer et al. (2021), tracked donor-reactive T cell clonotypes from pre- to post transplantation in N = 12 kidney transplant recipients with anti-CD25 induction [27]. In contrast to the other studies, these authors noted an increase in donor-reactive TRB diversity from pre- to post transplant timepoints, which varied from a week to 8 months post-transplant. 

Multiple studies have demonstrated deletion of donor-reactive CD4+ T cells in transplantation settings, but the mechanisms leading to this deletion have remained elusive. This study is the first to provide evidence that apoptosis is causing the deletion of donor-reactive CD4+ T cells post kidney transplantation. Next to this work and previous studies by our group [13,14], studies in murine models using apoptosis analyses and TCR-transgenic approaches also support a role of cell death in the peripheral deletion of donor-reactive T cells following transplantation [29,30,31]. A study by Wekerle et al. (2001) demonstrated that deletion of donor-reactive CD4+ T cells occurring early after bone marrow transplantation with costimulatory blockade (anti-CD154 plus CTLA4Ig) had features of both activation-induced cell death (Fas-dependent) and passive cell death (Bcl-xL-reversible) [32]. In addition, a study by Cippa et al. (2013), suggested that pro-apoptotic Bcl-2 family members played a critical role in the peripheral deletion of donor-reactive T cells [33].

We realize that CD137 expression probably does not capture all alloreactive T cells and in particular weakly activated T cells can be missed. However, in previous studies, we have shown that the CD137-expressing alloreactive T cells reflect the cells able to proliferate and the loss of polyfunctional CD137 T cells is closely associated with the development of DSH in time after transplantation [13,21]. In addition, high frequencies of CD137-expressing T cells were recently shown to be correlated with acute T cell-mediated rejection in kidney transplant recipients [34]. Therefore, these cells represent essential and clinically relevant alloreactive T cells in circulation. Indeed, this study observed differences in gene expression from pre- to post transplantation which were donor-specific and validated previous observations made on a protein level in CD137-expressing alloreactive T cells.

The method employed here has a few limitations. Firstly, the number of samples and cells sequenced per sample were low. The low number of cells sequenced per sample, in combination with the high diversity in the alloreactive TCR repertoire, limited the conclusions which could be drawn in respect to TCR clonality. Ideally, we would have included more recipients with paired donor-, as well as third-party-, stimulated samples from prior to and post transplantation with a greater number of alloreactive T cells sequenced. Secondly, clustering sorted T cells according to type (CD4+ and CD8+) and subset (e.g., naïve, central or effector T cells) is challenging using mRNA expression, as shown in the recent literature [35,36]. Despite the relatively low RNA expression levels of *CD4,* which were similar to those found in other studies on cryopreserved PBMCs, the high expression of *CD8* was sufficient to discriminate between the two [35,36]. However, we were unable to distinguish T cell differentiation status (i.e., naïve, central, effector and terminally differentiated memory T cells) based on RNA expression.

T cell subset discrimination could be greatly enhanced in future by combining protein and transcript analyses. This can be achieved either by sorting for subsets of interest before sequencing or using molecular cytometry, an adaptation of Next Generation Sequencing which simultaneously provides information about cellular transcripts and proteins. This has recently been applied in a study which compared the expression of 38 proteins and 399 target T cell transcripts at various timepoints following T cell activation [37]. This multi-omics approach would better allow us to discriminate between the CD4+ and CD8+ differentiation subsets for deeper profiling of classical T cell phenotypes.

In conclusion, the results from the single cell transcriptome and TRA/TRB repertoire sequencing of donor-reactive T cells before and after kidney transplantation supports the role of apoptosis of highly activated alloreactive CD4+ T cells while maintaining polyclonality.

## 4. Materials and Methods

### 4.1. Study Design

Heparinized peripheral blood samples of 7 stable kidney transplant recipients (P0, P1, P2, P5A, P5B, P5C, P5D; >54 years of age at time of transplantation) prior to and 3–5 years post kidney transplantation were used (Appendix A). A total of 10 donor-stimulated samples were included from prior to (N = 5; P0, P1, P2, P5A, P5B) and post transplantation (N = 5; P0, P1, P2, P5C, P5D). As a control, third-party-stimulated samples from a pre-transplant (N = 2; P0, P2) and post-transplant timepoint (N = 3; P0, P1, P2) were included. Criteria for participation included absence of any previous T cell depleting therapy and stable graft function with no signs of rejection. The immunosuppressive regimen consisted of basiliximab (Simulect^®^, Novartis, Basel, Switzerland) administered as non-depleting induction therapy followed by maintenance therapy with tacrolimus (Prograf^®^, Astellas Pharma) and mycophenolate mofetil (MMF) (Cellcept^®^, Roche, Basel, Switzerland) (Appendix A). The kidney transplant recipients included in this manuscript are part of the ongoing Gandalf study (project number: 18PhD08, funded by the Dutch Kidney Foundation). This study has been approved by the Medical Ethical Committee of the Erasmus Medical Centre (MEC No. 2018-048) and all kidney transplant recipients gave written informed consent to participate. This study adheres to the Declaration of Helsinki and the Declaration of Istanbul and is in conformance with the International Conference on Harmonization/Good Clinical Practice regulations.

### 4.2. PBMC Isolation 

Peripheral blood mononuclear cells (PBMCs) were isolated from heparinized peripheral blood samples on the day of blood sampling using Ficoll-Paque Plus (GE healthcare, Uppsala, Sweden) density centrifugation and stored at −150 °C with a minimum amount of 10 × 10^6^ cells per vial until further use, as described previously [38]. 

### 4.3. CD3+ T Cell Depletion of Allogeneic Stimuli

Recipient PBMCs were stimulated with CD3-depleted PBMCs from their respective kidney donor or a third party as control. Third-party candidates were selected based on being completely mismatched for the donor but having an equal number of HLA mismatches with the tested recipient as the donor. CD3 MACS^®^ MicroBeads (Miltenyi Biotec, Bergisch Gladbach, Germany) were used to deplete PBMCs of CD3+ T cells, according to manufacturer’s instruction. Flow cytometry was used to check for CD3+ T cell depletion efficiency (>95%). Before use, recipient PBMCs and CD3-depleted allogeneic stimulator cells were allowed to rest for 18 h at 37 °C. 

### 4.4. Sorting for CD137-Expressing Recipient T Cells

CD137-expressing T cells of stable kidney transplant recipients obtained prior to and at 3–5 years after transplantation were sorted following stimulation of 20 to 40 million recipient PBMCs with CD3-depleted allogeneic stimulator cells at a 1:0.5 ratio for 18–24 h. Stimulation was performed in polystyrene tubes (BD, Erembodegem, Belgium) in the presence of co-stimulation anti-CD49d (1  µg/mL; BD), as has been described in detail previously [21]. After stimulation, cells were washed with sterile PBS (without Mg^2+^ and Ca^2+^, pH 7.4, Invitrogen; Landsmeer, The Netherlands) at room temperature (RT). Cells were then surface stained using antibodies to sort the CD137-expressing T cells (Appendix A). Briefly, cells were stained for 15 min at RT with Fixable Viability Stain-780 (FVS780; BD) to exclude dead cells. Upon washing, cells were stained for 30 min at RT with APC-labelled CD137 antibody, BV510-labelled CD3 antibody and APC-H/Cy7-labeled antibodies directed to CD14, CD19 and CD56 (Appendix A). Cells were then washed and resuspended in PBS with 1% sterile heat-inactivated Fetal Calf Serum (HI-FCS) (Lonza, Verviers, Belgium) at a concentration of 20–25 × 10^6^ cells/mL. The cell suspension was filtered through a 35 µm nylon mesh using Falcon™ Round-Bottom Polystyrene Test Tubes with Cell-Strainer Snap (Thermo Fisher, Waltham, MA, USA). Cells were sorted using the BD FACSAria™ II Cell Sorter. A representative sample in Appendix A illustrates the gating strategy for sorting donor-reactive (CD137+CD3+DUMP-) T cells. Briefly, viable T cells were identified by gating lymphocytes using the forward and side scatter characteristics, after which singlets were identified using the side and forward scatter height and width parameters. The BV510 channel was used to identify CD3-expressing T cells and the APC-H/Cy7 was used to exclude unwanted cells positive for CD14 (monocytes), CD19 (B-cells), CD56 (natural killer cells) and FVS780 (dead cells). Viable CD3+ T cells expressing CD137 were sorted and collected into 15 mL falcon tubes coated with and containing 1 mL filtered HI-FCS. A small sample of the sorted cells was tested for purity (percentage of lymphocytes which are CD3+CD137+DUMP-) (Appendix A). Sorted cells were then washed in PBS 1% filtered HI-FCS and resuspended in degassed sterile PBS (without Mg^2+^ and Ca^2+^), pH 7.4 at RT at a volume of 350 µL and concentration of 2–5 × 10^4^ cells/mL for sequencing. Approximately 100–300 of the sorted alloreactive T cells were deposited per chip (Appendix A).

### 4.5. Combined Single Cell Transcriptomics and TRA/TRB Profiling

The ICELL8 Single-Cell System (Takara Bio, Shiga, Japan) was used for combined single-cell transcriptome and T Cell Receptor Alpha/Beta (TRA/TRB) profiling. This system has previously been validated for paucicellular samples. [39] In brief, cells were stained with Hoechst and propidium iodide (PI) using the Ready Probes Cell viability Imaging kit (Thermo Fisher Scientific, Waltham, MA, USA) and 160 µL at a concentration of 2–5 × 10^4^ cells/mL was dispensed per sample onto a blanco chip and pre-printed chip using the ICELL8 Single-Cell System. CellSelect automated microscopy image analysis software version 1.1.10.0. (Takara Bio) selected wells containing a single viable cell for in-chip full-length copy DNA (cDNA) amplification using SMARTScribe reverse transcriptase (Takara Bio). The resulting barcoded amplicons were purified off-chip using a NucleoSpin Gel and PCR Clean-up kit (Machery-Nagel, Düren, Germany) followed by an AMPure XP beads purification (Beckman Coulter, Brea, CA, USA). Quantification and quality control was performed on a Bioanalyzer using the High sensitivity DNA kit (Agilent Technologies, Santa Clara, CA, USA).

### 4.6. Amplification

Two sequencing libraries were generated from each single cell cDNA preparation, one for the TRA/TRB repertoire and one for the 5′ ends of transcripts, and, subsequently, sequenced as described previously [39]. Briefly, for transcriptome analysis, 5′ end transcriptome libraries were extended from the purified cDNA by a 12 cycle PCR using the Nextera XT DNA library prep (Illumina) and sequenced according to the Illumina TruSeq Rapid v2 protocol on Illumina HiSeq2500 sequencer. Single reads were generated of 50 bp in length with a single 8 bp index sequence. For TRA/TRB profiling, TRA and TRB transcripts were amplified using specific PCR reactions which also added the Illumina adapter sequences. From the resulting TR sequencing libraries, paired-end reads of 300 bp in length were generated with an 8 bp index on an Illumina MiSeq sequencer per the manufacturer’s instructions (Illumina, San Diego, CA, USA). 

### 4.7. Transcriptome and TCR Sequence Pre-Processing

Pre-processing of the single-cell sequenced RNA data was conducted as previously described [20]. In brief, reads (with well-barcodes and Illumina adapter sequences removed) were aligned to the GRCh38 human reference genome extended with exon-exon junctions using the HISAT2 aligner [39]. Resulting alignments were annotated with the well barcodes, converted to BAM format using SAMtools [40], and then converted to BED entries. These were intersected with Ensembl exons (release 96) [41] using BEDtools [42] and counts per gene were determined. For the TCR data, forward and reverse reads were merged using PEAR (version 0.9.2) [43]. The assembled sequences were then aligned to IMGT-defined TCR V, D and J reference sequences [44] using the IgBLAST tool (version 1.14) [45]. The resulting alignments were processed further in R (version 3.6.3).

### 4.8. Transcriptome Analysis

Single-cell analysis was performed in R using the Seurat single-cell analysis package version v3.1.4 [46]. Low quality or dying cells were removed from analysis by filtering out those cells with fewer than 500 expressed genes or those of which more than 5% of the total expression signal was from mitochondrial genes (Appendix A). Cells with more than 5000 genes likely resulted from doublets of cells instead of single cells and were also removed [46] (Appendix A). The ‘NormalizeData’ and ‘ScaleData’ functions from the Seurat package were used to normalize and scale expression signals, respectively. For normalization, feature counts for each cell were divided by the total counts for that cell, multiplied by the scale.factor (set to 10,000), and then natural-log transformed using log1p (returns the natural logarithm of counts plus 1). Data were corrected for batch effect between chips using scRNA-seq integration as described in Stuart et al., 2019 [31]. Dimensional reduction was performed using principal component analysis (PCA) based on the top 2000 most highly variable (protein coding) genes. Dimension reduction with the first 16 dimensions in the dataset was conducted using Uniform Manifold Approximation and Projection (UMAP). A resolution of 0.4 was chosen for clustering based on the clustree R package, available from CRAN (Appendix A) [47]. Differential gene expression analyses were performed using the raw RNA (unintegrated) scRNA-seq data for cells within each cluster compared to cells in all other clusters. We used the logistic regression differential expression test [48] implemented in the FindMarkers function in Seurat. We ranked the significant differentially expressed (DE) genes (adjusted *p*-value below 1 × 10^−5^ as determined by Seurat’s implementation of the Wilcoxon rank-sum test) according to average absolute log2 fold change (FC) in expression. We performed gene ontology (GO) enrichment analysis on this set of genes for both molecular function and biological process, using the R package “goseq” with an adjusted *p*-value cut-off of 1 × 10^−5^ [49]. A significant over-represented *p*-value indicated there are more DE genes in the category than we would expect given the size of the category and the gene length distribution. We repeated this differential gene expression analysis between samples from a pre- and post timepoint within each cluster. Only differential genes which were expressed in at least 5% of cells were retained. GO term analysis on DE genes was performed with an adjusted *p*-value cut-off of 0.1. Throughout the analysis, the tidyverse packages for R were used [50].

### 4.9. TCR Analysis

Sequencing data were obtained per locus (TRA, TRB and TRD). For cells where multiple loci were detected, the locus with the highest reads was selected per cell (r = 1). Only productive CDR3 sequences (in frame without an early stop codon and had a minimum of 25 reads per locus) were retained (Appendix A). In addition, only cells where the reads per locus was greater than 0.8 of the total reads for that locus (top = max(n)/reads) and where the fraction of reads per locus was greater than 0.10 (flocus = reads/sum(reads) >0.1) were retained. TCR analysis was performed on N = 3 renal recipients (P0, P1, P2) with paired donor-stimulated samples for both timepoints, of which two recipients (P0, P2) also had paired third-party-stimulated samples for both timepoints. 

### 4.10. Shannon Diversity Calculations 

The Shannon Diversity Index (*H*) was calculated for each TRA and TRB clonotype as well as the matched TRA AND TRB clonotypes of each sample. For each unique clonotype (*i*), the proportion $(p_{i}$) of cells in a sample with this clonotype was calculated using the following equation: $p_{i}=n_{i}/N$, where $n_{i}$ is the number of cells with clonotype *i* and *N* is the total amount of cells within a sample. *H* was then calculated for the total of unique clonotypes (*S*) using the following equation:$$H=-\sum_{i=1}^{S}\left(p_{i}*\ln\left(p_{i}\right)\right)$$

*H* was normalized for the total amount of unique clonotypes present within each sample (*S*), resulting in a Shannon Equitability Index (*EH*) ranging from 0 to 1.
EH=H/ln⁡(S)

An *EH* of 0 indicates that each cell has the same clonotype (low diversity), while an *EH* of 1 means all cells have a different clonotype (high diversity).

## Figures and Tables

**Figure 1 ijms-24-14463-f001:**
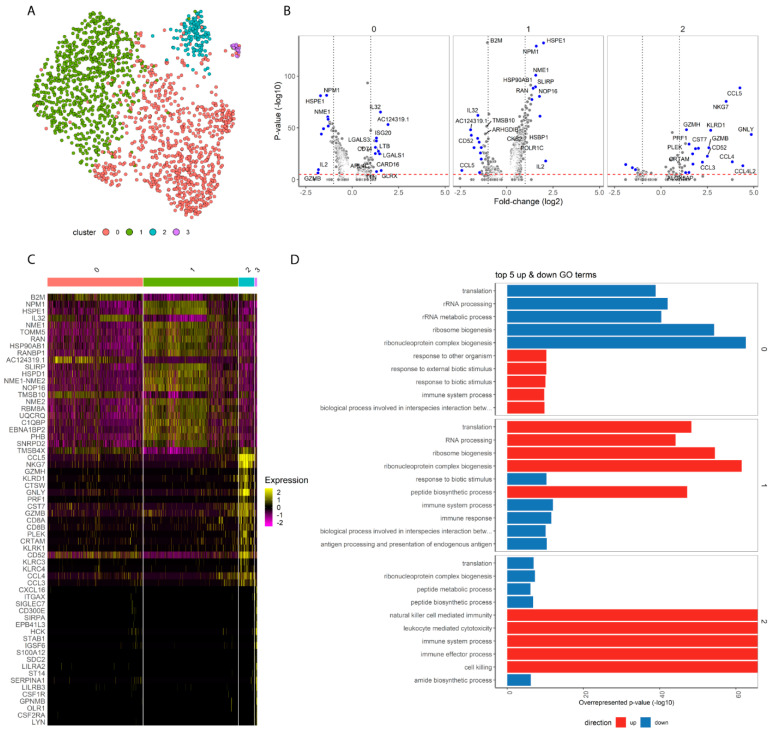
Differential gene analysis of alloreactive T cells reveals 2 clusters of CD4+ and 1 cluster of CD8+ T cells. (**A**) UMAP projections of the gene expression data of alloreactive T cells (N = 7 pre and N = 8 post) color-coded by cluster. (**B**) differential expression analysis of the cells in the cluster vs. the cells outside of the cluster. The log2 fold changes (*x*-axis) and adjusted *p*-values (*y*-axis) are depicted for the genes. A dashed line on the *x*-axis indicates log2 fold change of 1 and a line on the *y*-axis is set at adjusted *p*-value 1 × 10^−5^. The top 20 differential genes based on absolute log2 fold change with *p* < 0.00001 are depicted in blue. (**C**) heatmap illustrating the top 20 differentially regulated genes per cluster with purple indicating downregulation and yellow indicating upregulation relative to cells outside of the cluster. (**D**) top 5 up- and top 5 downregulated Gene Ontology (GO) terms based on all differentially expressed genes (adjusted *p* value < 0.01) between cells in a cluster vs. the cells outside of a cluster.

**Figure 2 ijms-24-14463-f002:**
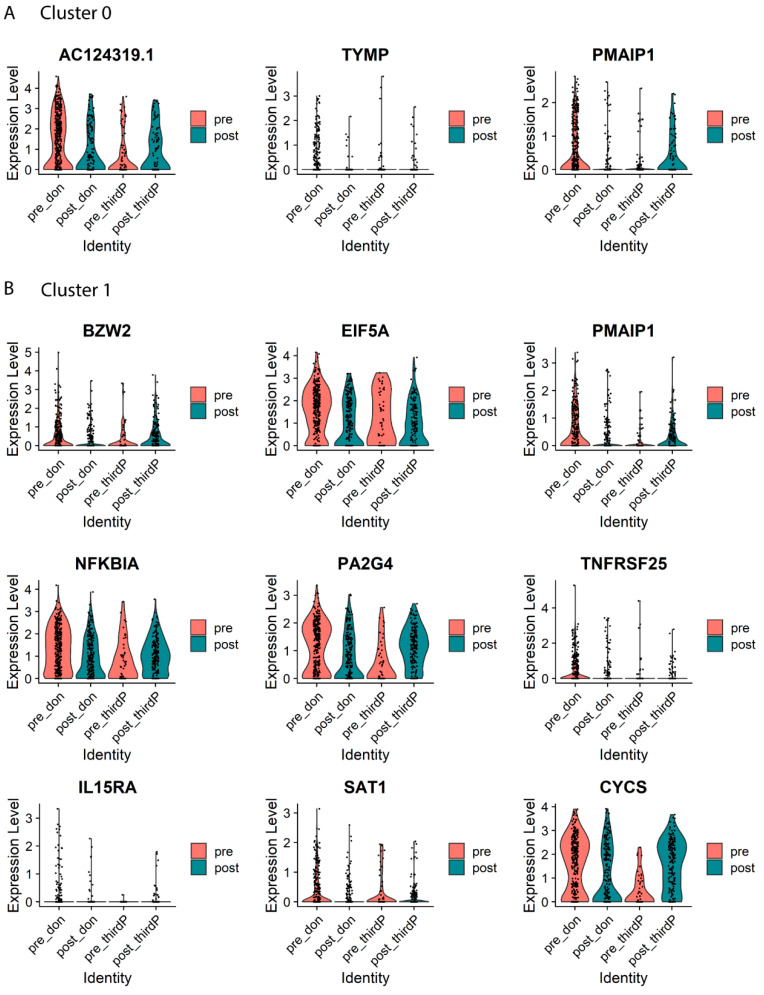
Violin plots of top differential genes between CD4 alloreactive T cells pre- and post-transplant. Violin plots depicting the expression level of the top differential genes reaching an adjusted p-value below 0.01 between samples from a post versus a pre-timepoint within clusters 0 (**A**) and cluster 1 (**B**) for donor-stimulated (N = 5 pre- and N = 5 post-transplant) and third-party-stimulated (N = 2 pre- and N = 3 post-transplant) samples. Cells are divided by stimulation type and timepoint. Cells from donor-stimulated samples from pre-transplantation and post-transplantation are present in the first two columns (pre_don, post_don) and third-party-stimulated samples from pre-transplantation and post-transplantation within the latter two columns (pre_thirdP, post_thirdP). The color of each column indicates whether cells are from a pre or post timepoint sample, with pre-transplant samples indicated in pink and post-transplant samples in blue.

**Figure 3 ijms-24-14463-f003:**
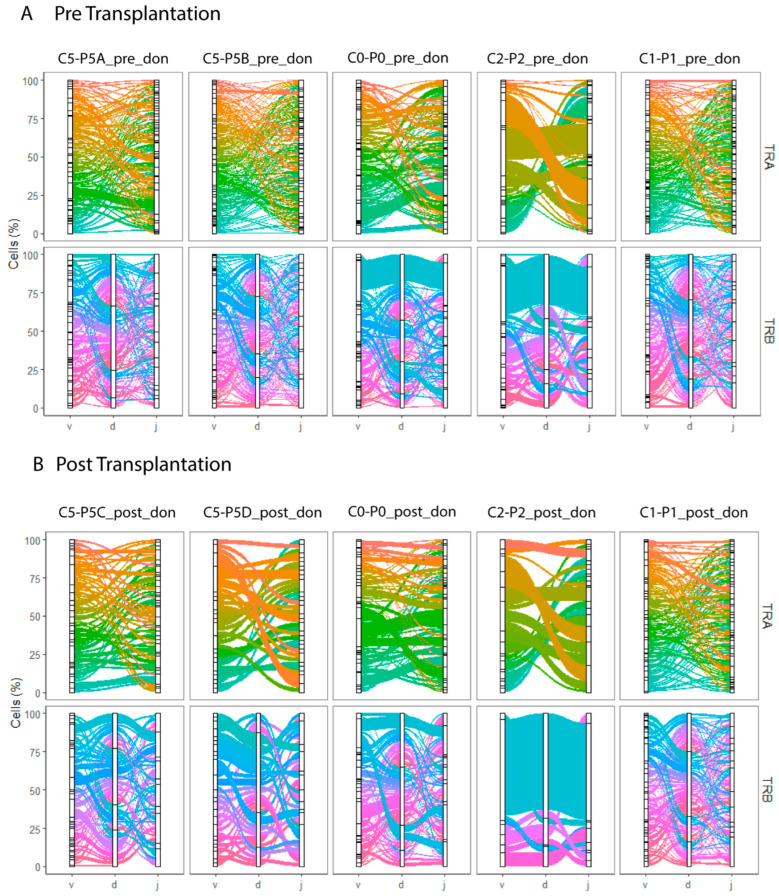
TRA and TRB clonotypes for samples containing donor-reactive T cells. Proportion of donor-reactive T cells (*y*-axis, %) from paired pre- and post-transplant samples of N = 5 kidney transplant recipients plotted against the V, D, J genes (*x*-axis) arranged as present in the human genome for TRA (top row) and TRB (bottom row) for samples from pre-transplant (**A**) and post-transplant (**B**). Colored lines differentiate unique V(D)J combinations, whereas the width of the line indicates the number of single cells with an identical V(D)J combination.

**Table 1 ijms-24-14463-t001:** Genes with differential expression (adjusted *p*-value < 0.01) between donor-stimulated alloreactive T cells within cluster 0 from before (pre_don) to 3–5 years after transplant (post_don).

Gene_ID	Symbol	Average log2FC	Adjusted *p*-Value	Description
ENSG00000173821	RNF213	−0.79	0.05	ring finger protein 213
ENSG00000025708	TYMP	−0.66	0.002	thymidine phosphorylase
ENSG00000141682	PMAIP1	−0.42	0.01	phorbol-12-myristate-13-acetate-induced protein 1

**Table 2 ijms-24-14463-t002:** Genes with differential expression (adjusted *p*-value < 0.01) between donor-stimulated alloreactive T cells within cluster 1 from before (pre_don) to 3–5 years after transplant (post_don).

Gene_ID	Symbol	Average log2FC	Adjusted *p*-Value	Description
ENSG00000136261	BZW2	−0.75	5.9 × 10^−3^	basic leucine zipper and W2 domains 2
ENSG00000132507	EIF5A	−0.73	4.8 × 10^−5^	eukaryotic translation initiation factor 5A
ENSG00000141682	PMAIP1	−0.61	7.1 × 10^−5^	Phorbol-12-myristate-13-acetate-induced protein 1
ENSG00000100906	NFKBIA	−0.57	1.1 × 10^−2^	NFKB inhibitor alpha
ENSG00000170515	PA2G4	−0.54	4.6 × 10^−4^	proliferation-associated 2G4
ENSG00000215788	TNFRSF25	−0.47	1.1 × 10^−2^	TNF receptor superfamily member 25
ENSG00000049249	TNFRSF9	−0.47	1.5 × 10^−2^	TNF receptor superfamily member 9
ENSG00000134470	IL15RA	−0.46	6.6 × 10^−3^	interleukin 15 receptor subunit alpha
ENSG00000130066	SAT1	−0.44	3.8 × 10^−2^	spermidine/spermine N1-acetyltransferase 1
ENSG00000172115	CYCS	−0.42	1.8 × 10^−2^	cytochrome c, somatic
ENSG00000136045	PWP1	−0.41	1.5 × 10^−2^	PWP1 homolog, endonuclein
ENSG00000134987	WDR36	−0.40	1.5 × 10^−2^	WD repeat domain 36
ENSG00000116717	GADD45A	−0.39	4.1 × 10^−3^	growth arrest and DNA damage inducible alpha
ENSG00000013306	SLC25A39	−0.38	3.1 × 10^−2^	solute carrier family 25 member 39
ENSG00000231925	TAPBP	−0.37	5.4 × 10^−3^	TAP binding protein
ENSG00000196396	PTPN1	−0.35	1.7 × 10^−3^	protein tyrosine phosphatase non-receptor type 1
ENSG00000143942	CHAC2	−0.33	4.1 × 10^−5^	ChaC cation transport regulator homolog 2
ENSG00000115946	PNO1	−0.28	3.1 × 10^−2^	partner of NOB1 homolog
ENSG00000105447	GRWD1	−0.27	9.9 × 10^−5^	glutamate rich WD repeat containing 1

**Table 3 ijms-24-14463-t003:** Top 10 differentially regulated GO Terms based on differentially expressed genes (adjusted *p*-value < 0.1) between samples from a pre- and post timepoint for cluster 1.

Category	Direction	Over- Represented *p*-Value	* DE Genes in Term	** Total Genes in Term	Term
GO:0033140	down	1.02 × 10^−5^	2	3	negative regulation of peptidyl-serine phosphorylation of STAT protein
GO:0033139	down	5.12 × 10^−5^	2	6	regulation of peptidyl-serine phosphorylation of STAT protein
GO:2001244	down	1.35 × 10^−4^	3	52	positive regulation of intrinsic apoptotic signaling pathway
GO:2001242	down	1.55 × 10^−4^	4	146	regulation of intrinsic apoptotic signaling pathway
GO:0042501	down	1.62 × 10^−4^	2	10	serine phosphorylation of STAT protein
GO:0019221	down	4.62 × 10^−4^	5	364	cytokine-mediated signaling pathway
GO:0033209	down	5.64 × 10^−4^	3	89	tumor necrosis factor-mediated signaling pathway
GO:0006364	down	7.43 × 10^−4^	4	219	rRNA processing
GO:0006915	down	7.84 × 10^−4^	9	1447	apoptotic process
GO:0033137	down	8.37 × 10^−4^	2	23	negative regulation of peptidyl-serine phosphorylation

* numDEInCat: number of differentially expressed genes in GO-term, ** numInCat: number of genes in GO Term.

**Table 4 ijms-24-14463-t004:** Shannon Equitability Index (*EH*) per sample for TRA, TRB and matched TRA AND TRB clonotypes.

Sample	TRA	TRB	TRA AND TRB
C0-P0_pre_don	0.92	0.87	0.99
C0-P0_post_don	0.90	0.92	0.98
C1-P1_pre_don	0.99	0.99	1.00
C1-P1_post_don	0.98	0.98	0.99
C2-P2_pre_don	0.78	0.79	0.94
C2-P2_post_don	0.94	0.69	0.93
C0-P0_pre_thirdP	0.87	0.92	0.97
C0-P0_post_thirdP	0.67	0.83	0.81
C2-P2_pre_thirdP	0.90	0.67	0.94
C2-P2_post_thirdP	0.78	0.61	0.79

## Data Availability

The datasets analyzed for this study can be found in the Gene Expression Omnibus under accession number GSE220956.

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
