# Peer review of "Single-Cell RNA Sequencing of Donor-Reactive T Cells Reveals Role of Apoptosis in Donor-Specific Hyporesponsiveness of Kidney Transplant Recipients"

_ijms, 2023, doi:10.3390/ijms241914463_

Round 1
Reviewer 1 Report
Amy C.J. van der List et al. presented the results of a single cell RNA and TCR seq of the alloreactive T cell to demonstrate the donor-specific hyporesponsiveness in T cell from kidney transplant patients.
The result is confusing, and more information need to be provided.
In section 1.1, the notion “In conclusion, gene expression changes from before to 3-5 years after transplantation in the TIGIT-expressing donor-reactive CD4+ T cells (cluster 0) show a downregulation in expression of inflammatory and apoptosis-related genes.”, the downregulated “inflammatory gene” is missing in the former part of the result.
Downregulation of apoptosis should be confirmed by phenotype or protein expression of the CD137+ alloreactive T cell after the MLR, only few gene expression is not enough to reach the conclusion.
Why the CD8+ alloreactive T cell in this study is so few? Providing the patients’ total Cd4 and CD8 T cell proportion before and after transplantation, and providing this information in the CD137+ cell population for donor antigen and third party antigen in each patient before FACS is helpful.
Why the most of the genes presented in figure2 are highly expressed by the pre-don than pre-thirdP? Is this caused by technical or statistical bias?
Please discuss the opposite trend of the pre to post transplant of some genes, such as the PMAIP1, BZW2, PMAIP1, NFKBIA, PA2G4, TNFRSF25, IL15RA, CYCS, between the don and thirdP group.
The table IV data possibly indicate the third party direct allo T cells are originate from clonally expanded clones. Please provide the D50 or Gini index to describe the clonality of the allo reactive clone. or please provide the T cell differentiation phenotype of the allo T cell before FACS. Please discuss the decrease of the EH of some third party clone after transplant than pre transplant.
Please provide the overlap of the third party TCR clone before and after transplant, and discuss if these clones are also turnover to the same extention as the don-T cell. And please provide the overlap between third party and don at each time point of each patients.
Author Response
Please see attachment for a revised version (with track changes) of the manuscript.
We would like to thank Reviewer 1 for their helpful comments. In response, we have provided additional information (please see point-by-point response below) and expanded on the results and discussion sections.
COMMENT 1: In section 1.1, the notion “In conclusion, gene expression changes from before to 3-5 years after transplantation in the TIGIT-expressing donor-reactive CD4+ T cells (cluster 0) show a downregulation in expression of inflammatory and apoptosis-related genes.”, the downregulated “inflammatory gene” is missing in the former part of the result.
In the results (previously section 1.1 – now section 3.4), we have removed the term “inflammatory” to describe the downregulation of genes in cluster 0 (TYMP, PMAIP1, and RNF231). We also removed the term “inflammatory” to describe the downregulation of genes in cluster 1 and instead focussed on their specific role – namely activation, proliferation and/or apoptosis.
COMMENT 2: Downregulation of apoptosis should be confirmed by phenotype or protein expression of the CD137+ alloreactive T cell after the MLR, only few gene expression is not enough to reach the conclusion.
Ideally the observed downregulation of apoptosis at the mRNA level would need to be confirmed at the protein level of the CD137+ alloreactive T cell after the MLR. This was not performed in this study as we only stained for CD3 and CD137 expression to FACS sort CD137-expressing T cells.
However, protein expression analysis for donor-reactive T cell phenotype and apoptosis has been performed using the same CD137 MLR assay in a separate article published by our group which we have referenced in our discussion (reference 14; van der List ACJ et al. J Immunol. 2022;209(7):1389-400). In this study, we demonstrated that CD4+ donor-reactive T cells which were highly active and secreted pro-inflammatory cytokines (IFN-gamma, TNF-alpha, IL-2) are present at reduced frequencies in the peripheral blood of stable kidney transplant recipients 3 years post-transplant. Therefore, we expected that the CD4+ donor-reactive T cells which remain in the periphery post-transplant have lower expression of mRNA related to T cell activation and cytokine expression. Indeed, this was observed in this study with downregulation of genes which play a role in cytokine-mediated signalling pathways.
Notably, the genes differentially regulated post-transplant within cluster 1 were also involved in the downregulation of apoptotic processes (GO:0006915, GO:2001244, GO:2001242) (Table III). In our previous publication, the proportion of donor-reactive CD137+ T cells undergoing apoptosis (positive for Annexin V) was significantly increased within the CD4+ but not the CD8+ subset post-transplantation for N=9 stable kidney transplant recipients 3-5 years after transplantation. Therefore, we hypothesized that the loss of CD4+ donor-reactive T cells was mediated through apoptosis and that those which remain in the periphery are the donor-reactive T cells more resilient to cell death through apoptosis. We have elaborated how multi-omics techniques could be employed to measure protein and mRNA expression within the same cells to prove this hypothesis in future research (section 4, page 11).
COMMENT 3: Why the CD8+ alloreactive T cell in this study is so few? Providing the patients’ total Cd4 and CD8 T cell proportion before and after transplantation, and providing this information in the CD137+ cell population for donor antigen and third party antigen in each patient before FACS is helpful.
The proportion of CD8+ alloreactive T cells measured in the study is relatively low compared to CD4+ alloreactive T cells. In this study we did not stain for CD4 or CD8 receptors before sorting, instead focussing on CD3 and CD137 to isolate the alloreactive T cells. Therefore, we were unable to measure the proportion of total CD4+ and CD8+ T cell proportion before and after transplantation. However, unpublished data from a previous study, characterizing alloreactive (CD137+) T cells at the protein level in 21 stable kidney transplant recipients (reference 14) revealed that the average percentage (STDEV) of CD4+ and CD8+ T cells within the total (CD3+) T cell compartment was 72 (11) % and 23 (10) %, respectively. These proportions did not differ between pre and post-transplant samples.
Following short-term stimulation with donor antigen, the total donor-reactive (CD137+) T cell compartment consisted of 87 (10) % CD4+ and 13 (10) % CD8 pre-transplant compared 82 (9) % CD4+ and 18 (9) % CD8+ post-transplant. Similarly the total third-party-reactive (CD137+) T cell compartment consisted 89 (7) % CD4+ and 11 (7) % CD8+ pre-transplant compared to 84 (16) % CD4+ and 16 (16) % CD8+ post-transplant.
The samples used for this paper were derived from the same cohort of kidney transplant recipients and we have added a sentence stating the predominance of CD4+ in alloreactive (CD137+) CD3+ T cells (section Results 3.1, page 6).
COMMENT 4: Why the most of the genes presented in figure 2 are highly expressed by the pre-don than pre-thirdP? Is this caused by technical or statistical bias?
In Figure 2, we would expect that the gene expression in ‘pre_don’ and ‘pre_thirdP’ samples to be similar as the recipient has not “seen” either. As indicated in the methods, third-party samples were selected based on having a similar number of mismatches to the acceptor as that of the donor to the acceptor – however, the type of mismatches are inherently different. It is possible that certain mismatches lead to a stronger or weaker alloreactive response, explaining the different expression patterns seen between donor and third-party stimulated samples pre-transplant. In addition, not all PBMCs taken pre-transplant from an acceptor had been stimulated by both a donor and third-party. Ideally we would like to have used paired samples of recipients prior to and post transplantation and compare both donor- as well as 3P-stimulated samples but we were limited in samples and only able to include 3 paired recipients prior to and post transplantation. This limitation is included within the discussion (section 4, page 10-11).
Importantly, the genes differentially expressed comparing pre to post transplantation for donor-stimulated samples are not differentially expressed for third party stimulated samples – indicating a donor-specific response.
COMMENT 5: Please discuss the opposite trend of the pre to post transplant of some genes, such as the PMAIP1, BZW2, PMAIP1, NFKBIA, PA2G4, TNFRSF25, IL15RA, CYCS, between the don and thirdP group.
That the expression of genes which were significantly downregulated in donor-stimulated samples post-transplant (i.e. PMAIP1, BZW2, NFKBIA, PA2G4, TNFRSF25, IL15RA, CYCS) appear to be upregulated in third-party samples is simply due to the lower number of cells present in ‘pre_thirdP’ samples. However, statistically the expression is not different between ‘pre_thirdP’ and ‘post_thirdP’ samples. This is indicated in Table S6, the genes which were differentially expressed pre to post transplant for donor-stimulated samples did not change significantly over time between third-party pre and post-transplant samples with the exception of CYCS which was differentially expressed (upregulated) from pre to post-transplant.
COMMENT 6: The table IV data possibly indicate the third party direct allo T cells are originate from clonally expanded clones. Please provide the D50 or Gini index to describe the clonality of the allo reactive clone. or please provide the T cell differentiation phenotype of the allo T cell before FACS. Please discuss the decrease of the EH of some third party clone after transplant than pre transplant.
As mentioned previously, we cannot provide the T cell differentiation phenotype of the samples in this manuscript as we only used CD3 and CD137 protein expression to identify alloreactive T cells for FACS sorting purposes. However, in our previous publication (reference 14; van der List ACJ et al. J Immunol. 2022;209(7):1389-400), we demonstrated a decrease in the proportion of central/effector memory alloreactive T cells post-transplantation. That combining protein and mRNA expression on alloreactive T cells for identification of T cell differentiation phenotype (naïve, central/effector memory) is needed in further research has been elaborated on in the discussion (section 4, page 11).
In reference to whether certain alloreactive T cell clones are expanded, the high diversity in the TCR repertoire combined with the low number of shared TCR clonotypes between samples (pre vs post or donor vs third-party), limits conclusions which can be drawn in respect to TCR clonality. This limitation has been added to the discussion (section 4, pages 9-11).
COMMENT 7: Please provide the overlap of the third party TCR clone before and after transplant, and discuss if these clones are also turnover to the same extension as the don-T cell. And please provide the overlap between third party and don at each time point of each patients.
As described in the discussion, the high diversity in the TCR repertoire and the low number of cells sequenced lead to too few cells with a shared TRA or TRB clonotype between samples (either between timepoints or stimulations) for analysis.
The number of cells with a shared TRA or TRB clonotype between pre to post transplantation for donor-stimulated samples was already included in the manuscript in Supplementary Table 8. This table demonstrates that the average number of cells which expressed shared TRA or TRB clonotypes from pre to post-transplant averaged 1. Third-party samples had even fewer cells within a sample and, therefore, no conclusions on TCR overlap could be drawn based on so few cells. Instead, we have added the limitations of the cell numbers to this type of analysis to the discussion (section 4, page 9-11).

Reviewer 2 Report
This paper shows nicely the transcriptional properties of alloreactive T cells over time in kidney transplant recipients using single cell RNA sequencing. The donor-specific hyporesponsiveness of alloreactive T cells in kidney transplant recipients is explained by the loss of pro-apoptotic and activated alloreactive T cells over time as the alloreactive T cells found in the periphery of stable kidney transplant recipients are lacking these transcriptomic profiles. These data support previous studies propose a deletion/loss of activated alloreactive T cells.
-Patient recruitment ist not intuitive and number of analized samples should be stated at every figure as some analysis are performed on as few as 2 samples (n=2).
-Labeling on the figures should be improved for better understanding.
-Regarding a depletion efficiency of 95% of donor T cells prior to the experiments, how can it be excluded that antigen recognizing and therefore proliferating donor/third party T cells are captured in further analysis.
-The small sample size and clone counts should be adressed in the discussion.
-Headings numbering of the results section should be corrected.
-The second paragraphe of the second result section 1.1 (sic!) should be moved to the discussion (last sentence with references). Similarly the 2nd paragraph containing references in the 4th result section 1.1 (sic!) should be transferred to the discussion.
-How do the authors comment on the value of diversity calculations on clone numbers as low as 35 cells for TRA, 40 cells for TRB and 1 for TRD (suppl table III)
-The heterogeneous immunosuppression of the included patients should be discussed (15% mTOR only, 30% CNI only, 60% CNI+MMA) and which samples are what patients. Is there a difference in the TCR properties in these patients?
The Abbreviation TR should be explained.
"T Cell Receptor", "pre-transplant, post-transplant" should be written the same way throughout the manuscript.
The last three words of the discussion should be removed ("Conflict of interest")
Author Response
Please see attachment for the revised version (with track changes) of the manuscript.
COMMENT 1: This paper shows nicely the transcriptional properties of alloreactive T cells over time in kidney transplant recipients using single cell RNA sequencing. The donor-specific hyporesponsiveness of alloreactive T cells in kidney transplant recipients is explained by the loss of pro-apoptotic and activated alloreactive T cells over time as the alloreactive T cells found in the periphery of stable kidney transplant recipients are lacking these transcriptomic profiles. These data support previous studies propose a deletion/loss of activated alloreactive T cells.
We would like to thank Reviewer 2 for their helpful comments.
COMMENT 2: Patient recruitment is not intuitive and number of analysed samples should be stated at every figure as some analysis are performed on as few as 2 samples (n=2).
We have added the number of analysed samples to the caption of every main figure.
COMMENT 3: Labelling on the figures should be improved for better understanding.
We have improved the labelling on the figures within the caption for better understanding.
COMMENT 4: Regarding a depletion efficiency of 95% of donor T cells prior to the experiments, how can it be excluded that antigen recognizing and therefore proliferating donor/third party T cells are captured in further analysis.
We cannot exclude that some allogeneic (donor or third-party-derived) T cells have been sequenced in our analysis. However, as stated in the methods, recipient PBMCs are stimulated with CD3-depleted allogeneic stimulator cells at a 1:0.5 ratio. Therefore, possible contamination with 5% CD3-expressing donor or 3P-derived T cells is limited and unlikely to have an influence on the sequencing results.
COMMENT 5: The small sample size and clone counts should be addressed in the discussion.
We have addressed the small sample size and clone counts within the limitation section of the discussion (section 4, page 11).
COMMENT 6: Headings numbering of the results section should be corrected.
We have corrected heading numbering of the results section.
COMMENT 7: The second paragraph of the second result section 1.1 (sic!) should be moved to the discussion (last sentence with references). Similarly the 2nd paragraph containing references in the 4th result section 1.1 (sic!) should be transferred to the discussion.
We have moved the indicated sentences containing references from the results (section 3) to the discussion (section 4), as requested.
COMMENT 8: How do the authors comment on the value of diversity calculations on clone numbers as low as 35 cells for TRA, 40 cells for TRB and 1 for TRD (suppl table III)
We agree with the reviewer that the number of cells with TRD data were too low for diversity calculations. In the case of TRA and TRB – the sample numbers were sufficient to gain an indication of how diverse the pool of alloreactive T cells is and remains post-transplantation. However, we agree with the reviewer that no conclusions on changes on specific TCR clones can be made based on such low numbers. We have emphasized this limitation of small sample size for TCR clonality calculations in our discussion (section 4, page 9-11).
COMMENT 9: The heterogeneous immunosuppression of the included patients should be discussed (15% mTOR only, 30% CNI only, 60% CNI+MMA) and which samples are what patients. Is there a difference in the TCR properties in these patients?
The number of patients analysed are too few to make concrete associations between type of immunosuppression and RNA expression of alloreactive T cells. Moreover, that apoptosis-related genes are only downregulated post-transplantation in the donor-stimulated samples and not in the third-party-stimulated samples indicates a donor-specific effect separate from immunosuppression.

Round 2
Reviewer 1 Report
no comment
Reviewer 2 Report
The authors have adressed the major limitations (sample size and further conclusions) in the discussion